# Accumulation of Plastic Strain at Notch Root of Steel Specimens Undergoing Asymmetric Fatigue Cycles: Analysis and Simulation

**DOI:** 10.3390/ma16062153

**Published:** 2023-03-07

**Authors:** Faezeh Hatami, Ahmad Varvani-Farahani

**Affiliations:** Department of Mechanical and Industrial Engineering, Toronto Metropolitan University, Toronto, ON M5B 2K3, Canada

**Keywords:** local ratcheting, A-V kinematic hardening model, backstress evolution, neuber, Hoffman-Seeger, Glinka rule, finite element analysis, Chaboche’s model

## Abstract

The present study evaluates the ratcheting response at notch roots of 1045 steel specimens experiencing uniaxial asymmetric fatigue cycles. Local stress and strain components at the notch root were analytically evaluated through the use of Neuber, Glinka, and Hoffman-Seeger (H-S) rules coupled with the Ahmadzadeh-Varvani (A-V) kinematic hardening model. Backstress promotion through coupled kinematic hardening model with the Hoffman-Seeger, Neuber, and Glinka rules was studied. Relaxation in local stresses on the notched samples as hysteresis loops moved forward with plastic strain accumulation during asymmetric loading cycles was observed. Local ratcheting results were simulated through FE analysis, where the Chaboche model was employed as the materials hardening rule. A consistent response of the ratcheting values was evidenced as predicted, and simulated results were compared with the measured ratcheting data.

## 1. Introduction

In the presence of stress raisers, load-bearing components are vulnerable to catastrophic failure, especially when they are subjected to asymmetric stress cycles in which the local stress state exceeds the elastic limit resulting in plastic strain accumulation referred to as local ratcheting. The presence of stress raisers intensifies the ratcheting progress during loading cycles. Investigations on how the ratcheting progress at the notch root behaves and how local stresses at the notch root relax out have been the center of attention for researchers [1,2,3,4]. Wang et al. [3] investigated steel specimens subjected to asymmetric loading cycles. They proposed an integral approach to define the plastic shakedown rate as the loading cycles proceeded. Hu et al. [4] reported that, as the applied strain increased, the local ratcheting and stress relaxation rate was further promoted at the notch root. Their measured ratcheting data and stress relaxation at the notch roots were reported in consistent agreement with those predicted by means of the Chaboche hardening rule. Rahman et al. [5] performed ratcheting tests on notched 304L steel plates with various notch geometries/shapes. They employed Chaboche, Ohno-Wang, and AbdelKarim-Ohno hardening rules to evaluate the ratcheting response of notched specimens. The ratcheting response and mean stress relaxation of S32750 steel bars were examined in a paper by Lee et al. [6] under the step-loading spectra. They developed a constitutive model to predict the mean stress relaxation and ratcheting, whose accuracy was confirmed through uniaxial loading tests. Strains were measured at the notch root of 1070 steel specimens undergoing axial-torsional loading cycles by Firat [7]. He employed the Chaboche model and Neuber’s rule to evaluate progressive plastic strains over cycles. The predicted local ratcheting strain values at the notch root of 1070 steel specimens were found in close agreement with those measured values reported in reference [8]. Kolahsangiani and Shekarian [9,10] examined local ratcheting at the notch root of 1045 steel plate specimens with various circular notch sizes. They discussed the influence of the notch size and stress levels on the local ratcheting magnitude and rate as the stress cycles increased. The coupled A-V hardening model [1] with the Neuber rule [11] was utilized to study the local ratcheting and stress relaxation at the notch vicinity of 1045 steel specimens [12]. The coupled framework governed the progressive plastic strain and stress relaxation at the notch root concurrently. Liu et al. [13] carried out some experimental tests with a high-stress concentration on Z2CND18.12N austenitic stainless steel elbow pipes. The measured local strains through strain gauges mounted on the circumference of pressurized elbow pipes closely agreed with the predicted ratcheting by means of the Chen-Jiao-Kim (CJK) model [14]. Ratcheting progress over loading cycles as well as stress relaxation at the notch roots of steel plates at constant strain ranges, was evaluated by Shekarian et al. [15]. They coupled the kinematic hardening rules of Chaboche and A-V with the Neuber rule to estimate local ratcheting at the notch roots. They found a close agreement between the predicted and measured values at various notch sizes and load levels. In another study, ratcheting at the roots of different elliptical and circular notches in 316 stainless steel specimens [16] was evaluated by means of the A-V and Chaboche hardening models in conjunction with the Neuber rule. Local ratcheting results were discussed on the basis of the choice of hardening models and their related influential parameters.

The present study evaluated the measured local ratcheting response of 1045 steel specimens [17,18,19] at the notch root employing the A-V hardening rule coupled with the Neuber, Glinka, and H-S models. Local ratcheting strains at the notch roots were calculated through the hardening framework during asymmetric loading cycles. Backstress evolution through the A-V model directly affected the local ratcheting rate and magnitude as different models were employed. The H-S model with equivalent stress terms presented a slightly sharper drop in stress at the notch root as loading cycles were applied. This model, however, resulted in lower ratcheting as coupled with the A-V hardening model. The corresponding backstress terms in this model affected the ratcheting rate and magnitude as the H-S rule was coupled in the framework. Numerically determined local ratcheting strains at notch root by means of the FE analysis fell below the experimental and predicted results. Simulated results were affected by the FE element size and types taken at different distances at the notch root vicinity and their related convergence. The choice of models in assessing the local strain components and their terms/constants was found to affect the rate and magnitude of ratcheting and stress relaxation during stress cycles.

## 2. Modeling and Formulation

### 2.1. Elastic and Plastic Strains

The total strain increment tensor was determined through the summation of elastic and plastic strain increments tensors as:(1)$$d\bar{\varepsilon }=d{\bar{\varepsilon }}_{e}+d{\bar{\varepsilon }}_{p}$$

While the elastic strain increment tensor was determined by Hooke’s law as:(2)$$d{\bar{\varepsilon }}_{e}=\frac{d\bar{\sigma }}{2G}-\frac{\vartheta }{E}\left(d\bar{\sigma }.\;\bar{I}\right)\bar{I}$$
where *E* is the modulus of elasticity, *G* is the shear modulus, ϑ is Poisson’s ratio, and terms $\bar{I}$ and $\bar{\sigma }$ correspond to unit and stress tensors, respectively.

Based on the associated flow rule, the plastic strain increment tensor can be defined as:(3)$$d{\bar{\varepsilon }}_{p}=\frac{1}{H_{p}}\left(d\bar{s}.\;\bar{n}\right)\bar{n}$$
where $H_{p}$ is the plastic modulus, $d\bar{s}$ is the increment of the deviatoric tensor, and $\bar{n}\;$corresponds to the normal vector of the yield surface. The yield criterion represents the onset of yielding where the yield contour separates the elastic domain from the plastic region through:(4)$$f\left(\bar{s},\alpha ,\sigma_{y}\right)=\frac{3}{2}\left(\bar{s}-\bar{\alpha }\right)\left(\bar{s}-\bar{\alpha }\right)-\sigma_{y}^{2}$$

In Equation (4), $\bar{\alpha }$ corresponds to the backstress tensor, translating the yield surface within the deviatoric stress space as the loading exceeds an elastic domain.

### 2.2. The Ahmadzadeh-Varvani (A-V) Kinematic Hardening Rule

During plastic deformation, the movement direction of the yield surface in the stress space is governed by the kinematic hardening rule. The A-V nonlinear hardening model [1] was structured to control the evolution of backstress increments over the loading process. The general form of the A-V rule is given as:(5a)$$d\bar{\alpha }=Cd{\bar{\varepsilon }}_{p}-\gamma_{1}\left(\bar{\alpha }-\delta \bar{b}\right)dp$$
(5b)$$d\bar{b}=\gamma_{2}\left(\bar{\alpha }-\bar{b}\right)dp$$

The first term of the A-V model corresponds to strain hardening, and the second term presents the dynamic recovery term to accommodate plastic strain accumulation. The internal variable, $\bar{b}$, with a zero initial value, was introduced to the dynamic recovery term of the hardening rule to gradually control backstress $\bar{\alpha }$ over loading cycles. Details of how to determine variables in Equations (5a) and (5b) were given in Ref. [1]. In Equations (5a) and (5b),dp is defined through the dot product of plastic strain increment $d{\bar{\varepsilon }}_{p}$ as:(6)$$dp=\sqrt{d{\bar{\varepsilon }}_{p}.d{\bar{\varepsilon }}_{p}}$$

Coefficients C and $\gamma_{1}$ in Equation (5) are defined from uniaxial stress-strain hysteresis loops. Constants $\gamma_{2}$ and δ are material-dependent coefficients [17]. For uniaxial loading conditions, δ is defined as ${\left(\bar{\alpha }/k\right)}^{m}$ expanding Equation (5a) to:(7)$$d\bar{\alpha }=Cd{\bar{\varepsilon }}_{p}-\gamma_{1}\left(\bar{\alpha }-{\left(\bar{\alpha }/k\right)}^{m}\;\bar{b}\right)dp$$where the coefficient *k* is defined as $k=C/\gamma_{1}$. Exponent m is material dependent and stays less than the unity 0<m<1.0.

### 2.3. Local Components of Stress and Strain at Notch Root and Local Ratcheting Strains

#### 2.3.1. Neuber’s Rule

Neuber’s rule relates the stress and strain concentration factors, $K_{\sigma }$, and $K_{\varepsilon }$ to the theoretical stress concentration factor, $K_{t}$ through:(8)$$K_{\sigma }K_{\varepsilon }=K_{t}^{2}$$
where:(9)$$K_{\sigma }=\frac{\sigma }{S}$$
(10)$$K_{\varepsilon }=\frac{\varepsilon }{e}$$

Substituting Equations (9) and (10) into Equation (8) resulted in:(11)$${\left(K_{t}S\right)}^{2}=E\sigma \varepsilon$$

For the cyclic stress and strain ranges, Equation (11) is rewritten as:(12)$$\frac{{\left(K_{t}\Delta S\right)}^{2}}{E}=\Delta \sigma \Delta \varepsilon$$
where *S* is the nominal stress, *e* is the nominal strain, *E* is the modulus of elasticity, σ, and ε are the local stress and strain at the notch root, respectively. To predict the local stress and strain at the notch root, Neuber’s rule [11] was developed for plane stress conditions as [9,10,12,15,16,18]:(13)$$\left(\varepsilon_{BL}-\varepsilon_{AL}\right)\left(\sigma_{BL}-\sigma_{AL}\right)=K_{t}^{2}\left(S_{B}-S_{A}\right)\left(e_{B}-e_{A}\right)\;\;d{\bar{\varepsilon }}_{p}<0$$
(14)$$\left(\varepsilon_{CL}-\varepsilon_{BL}\right)\left(\sigma_{CL}-\sigma_{BL}\right)=K_{t}^{2}\left(S_{C}-S_{B}\right)\left(e_{C}-e_{B}\right)\;\;d{\bar{\varepsilon }}_{p}\geq 0$$
where subscripts *A, B, C* correspond to the loading turning points starting from zero to the maximum load (point *A*), minimum load (point *B*), and maximum load (point *C*), and $K_{t}$ is the stress concentration factor. The uniaxial nominal strain and stress range are related through the Ramberg-Osgood equation as:(15)$$\Delta e=\frac{\Delta S}{E}+2{\left(\frac{\Delta S}{2K\prime }\right)}^{\frac{1}{n\prime }}$$
where K′ and n′ correspond to the cyclic hardening coefficient and exponent, respectively.

Substituting Equation (15) into Equations (13) and (14) resulted in Equations (16) and (17) being related to the local strain and stress components as [9,10,12,15,16,18]:(16)$$\left(\varepsilon_{B}-\varepsilon_{A}\right)\left(\sigma_{B}-\sigma_{A}\right)=K_{t}^{2}\left(S_{B}-S_{A}\right)\left(\frac{S_{B}-S_{A}}{E}+2{\left(\frac{S_{B}-S_{A}}{2K\prime }\right)}^{\frac{1}{n\prime }}\right),\;\;d{\bar{\varepsilon }}_{p}<0$$
(17)$$\left(\varepsilon_{C}-\varepsilon_{B}\right)\left(\sigma_{C}-\sigma_{B}\right)=K_{t}^{2}\left(S_{C}-S_{B}\right)\left(\frac{S_{C}-S_{B}}{E}+2{\left(\frac{S_{C}-S_{B}}{2K\prime }\right)}^{\frac{1}{n\prime }}\right),\;\;d{\bar{\varepsilon }}_{p}\geq 0$$

For components subjected to uniaxial loading, the local backstress component for loading the half-cycle could be related to local stress at turning points [3]. It can be used for unloading (A→B) and reloading conditions (B→C) through Equations (18)–(20):(18)$$\alpha_{AL}=\frac{2}{3}\left(\sigma_{AL}-\sigma_{y}\right)$$
(19)$$\alpha_{BL}=\frac{2}{3}\left(\sigma_{BL}+\sigma_{y}\right)$$
(20)$$\alpha_{CL}=\frac{2}{3}\left(\sigma_{CL}-\sigma_{y}\right)$$

#### 2.3.2. Glinka’s Rule

Molski and Glinka [20] presented an alternative to Neuber’s rule, which was based on the equivalent strain energy density (ESED). This method took the strain energy density at the notch root equal to a condition at which the loaded specimen stayed within the elastic domain. They attributed the stress concentration factor, $K_{t}$ to the strain energy through:(21)$$K_{t}=\frac{\sigma }{S}={\left(\frac{W_{\sigma }}{W_{S}}\right)}^{1/2}$$
where terms $W_{S}$ and $W_{\sigma }$ corresponded to the elastic strain energy per unit volume due to the nominal remote stress *S* and the strain energy per unit volume due to the local strain and stress at the notch root, respectively. For the notched specimen, the elastic and total strain energy per unit volume were determined by:(22)$$W_{S}=\frac{1}{2}S.e=S^{2}/2E$$
and
(23)$$W_{\sigma }=\int_{0}^{\varepsilon }\sigma \left(\varepsilon \right)d\varepsilon =\frac{\sigma^{2}}{2E}+\frac{\sigma }{n^{\prime }+1}{\left(\frac{\sigma }{K^{\prime }}\right)}^{\frac{1}{n^{\prime }}}$$
where, through Hooke’s law, the elastic strain and stress is related by $\sigma \left(\varepsilon \right)=E\varepsilon$. By replacing terms $W_{S}$ and $W_{\sigma }$ from Equations (22) and (23) with Equation (21), the stress concentration factor $K_{t}$ could be rewritten as:(24)$$K_{t}={\left\{\frac{\left(\frac{\sigma^{2}}{2E}+\frac{\sigma }{n\prime +1}{\left(\frac{\sigma }{K\prime }\right)}^{\frac{1}{n\prime }}\right)}{\frac{S^{2}}{2E}}\right\}}^{1/2}$$

Through the use of Equation (24), the relationship between the applied and local stress ranges could be determined as:(25)$$\frac{{\left(K_{t}\Delta S\right)}^{2}}{4E}=\frac{\Delta \sigma^{2}}{4E}+\frac{\Delta \sigma }{n\prime +1}{\left(\frac{\Delta \sigma }{2K\prime }\right)}^{\frac{1}{n\prime }}$$

To relate the stress and strain terms, the Ramberg-Osgood equation was adapted along with Equation (25) as:(26)$$\Delta \varepsilon \left(2E\left(\Delta \sigma \right)\right)+\left(n\prime -1\right)\Delta \sigma^{2}=K_{t}^{2}{\left(\Delta S\right)}^{2}\left(n\prime +1\right)$$

Considering subscripts turning points A, B, C over A→B and B→C loading paths, Equation (26) was expended as:(27)$$\left(\varepsilon_{B}-\varepsilon_{A}\right)\left(2E\left(\sigma_{B}-\sigma_{A}\right)\right)+\left(n\prime -1\right){\left(\sigma_{B}-\sigma_{A}\right)}^{2}=K_{t}^{2}{\left(S_{B}-S_{A}\right)}^{2}\left(n\prime +1\right)\;\;d{\bar{\varepsilon }}_{p}<0$$
(28)$$\left(\varepsilon_{C}-\varepsilon_{B}\right)\left(2E\left(\sigma_{C}-\sigma_{B}\right)\right)+\left(n\prime -1\right){\left(\sigma_{C}-\sigma_{B}\right)}^{2}=K_{t}^{2}{\left(S_{C}-S_{B}\right)}^{2}\left(n\prime +1\right)\;\;{\bar{\varepsilon }}_{p}\geq 0$$

#### 2.3.3. Hoffman and Seeger (H-S) Approach

Hoffman and Seeger (H-S) [21,22] proposed a method to establish a load-equivalent notch stress and strain relationship. The H-S method consisted of two steps: (i) uniaxial quantities σ, ε, and $K_{t}$ were initially replaced by the equivalent quantities ($\sigma_{q},\;\varepsilon_{q},$ and $K_{tq}$) on the basis of the von-Mises yield criterion; (ii) the equivalent values were related to the principal stress and strain components at the notch root. Within the elastic limit, the general form of the H-S model was presented as:(29)$$\varepsilon_{q}=\frac{\sigma_{q}}{E}F\left(\frac{\sigma_{e,q}}{\sigma_{q}}\right)$$
where $\varepsilon_{q}$ and $\sigma_{q}$ respectively correspond to the equivalent strain and stress components at notch root. Function $F\left(\frac{\sigma_{e,q}}{\sigma_{q}}\right)$ falls between $1\leq \frac{\sigma_{e,q}}{\sigma_{q}}<K_{p}$. Term *K_p_* corresponds to the limit load factor, which is the ratio of the ultimate load *L_p_* to the yield initiation load *L_y_* for elastic-perfectly plastic material.
(30)$$\;K_{p}=\frac{L_{p}}{L_{y}}$$

Using the von-Mises yield criterion, the theoretical elastic equivalent stress at notch root, $\sigma_{eq}$, can be defined as:(31)$$\sigma_{eq}=\sigma_{e1}\sqrt{\frac{1}{2}\left[{\left(1-a_{e}\right)}^{2}+{\left(1-b_{e}\right)}^{2}+{\left(a_{e}-b_{e}\right)}^{2}\right]}$$

Subscripts 1, 2, and 3 denote principal stress directions, and subscripts *e* represent stress levels within the elastic domain. Stress ratios within the elastic domain $a_{e}$ and $b_{e}$ are defined as:(32)$$a_{e}=\frac{\sigma_{e2}}{\sigma_{e1}}$$
(33)$$b_{e}=\frac{\sigma_{e3}}{\sigma_{e1}}$$

The equivalent stress concentration $K_{tq}$ is defined as:(34)$$K_{tq}=\frac{\sigma_{eq}}{S}$$

The relationship between $K_{tq}$ and $K_{t}$ is given as [21]:(35)$$K_{tq}=K_{t}\sqrt{\frac{1}{2}\left[{\left(1-a_{e}\right)}^{2}+{\left(1-b_{e}\right)}^{2}+{\left(a_{e}-b_{e}\right)}^{2}\right]}$$

The H-S approach [21] was developed based on the equivalent strain at notch root through:(36)$$\varepsilon_{q}=\frac{K_{tq}^{2}S^{2}}{\sigma_{q}}\frac{e^{*}}{S^{*}}$$
where $\sigma_{q}$ is the equivalent stress obtained through the von-Mises yield criterion. Terms $S^{*}$ and $e^{*}$ are respectively defined as $S^{*}=\left(K_{tq}/K_{p}\right)S$ and e*=σy/ES*/σy1n.

The equivalent applied stress and strain terms are then related to the local components over the unloading (A→B) path and reloading (B→C) path for each stress cycle through Equations (37) and (38), respectively.
(37)$$(\varepsilon_{qB}-\varepsilon_{qA})\left(\sigma_{qB}-\sigma_{qA}\right)=\frac{K_{tq}^{2}\left(S_{B}-S_{A}\right)}{\left(S_{B}^{*}-S_{A}^{*}\right)}\left(e_{B}^{*}-e_{A}^{*}\right)\;\;d{\bar{\varepsilon }}_{p}<0$$
(38)$$(\varepsilon_{qC}-\varepsilon_{qB})\left(\sigma_{qC}-\sigma_{qB}\right)=\frac{K_{tq}^{2}\left(S_{C}-S_{B}\right)}{\left(S_{C}^{*}-S_{B}^{*}\right)}\left(e_{C}^{*}-e_{B}^{*}\right)\;\;d{\bar{\varepsilon }}_{p}\geq 0$$

### 2.4. Ratcheting Analysis Algorithm

An algorithm was developed to predict the ratcheting and stress relaxation of notched 1045 steel specimens through the coupled hardening framework. Through the A-V hardening rule, the yield surface evolution was controlled within the plastic domain and backstress term $\left(\bar{a}-\delta \bar{b}\right)$ and dropped up to a steady-state condition as asymmetric loading cycles progressed. To evaluate local ratcheting, local cyclic stress and strain components at the notch root were calculated by coupling Neuber, Glinka, and H-S rules to the hardening framework. The algorithm program enabled an assessment of local ratcheting at constant stress cycles, and stress relaxation was monitored over asymmetric loading cycles at a given constant strain. Backstress $\bar{\alpha }$ and internal variable $\bar{b}$ were to control the increments of plastic strain $d{\bar{\varepsilon }}_{p}$. The algorithm to run the ratcheting program through the hardening rule framework was developed through a number of steps:

(i)Applied cyclic stresses to the notched specimens were introduced into the program,(ii)Through Equations (5a) and (5b), the backstress component $\bar{\alpha }$, internal variable $\bar{b}$, and term $\left(\bar{a}-\delta \bar{b}\right)$ were related to plastic strain increments over the loading progress,(iii)The plastic strain increment, $d{\bar{\varepsilon }}_{p}$, was computed through (i) Equations (16) and (17) based on Neuber’s rule, (ii) Equations (27) and (28) based on the Glinka approach, and (iii) Equations (37) and (38) by means of the H-S model.(iv)The accumulation of the progressive local plastic strain at the notch root, $d{\bar{\varepsilon }}_{p}$ was controlled through the A-V hardening model while Neuber, Glinka, and H-S rules were coupled to the hardening framework.(v)Through Equations (18)–(20), the backstress components were defined during unloading (*A*→*B*)/reloading (*B*→*C*) paths and set as equal to their counterpart increments in Equations (5a)–(7). This enabled us to set relationships between nominal and local stress components in the coupled framework.(vi)The ratcheting strain was calculated from the average of maximum and minimum local strains over asymmetric loading cycles.

## 3. Testing Conditions and Ratcheting Data

Ratcheting data sets on notched 1045 steel specimens were taken from an earlier article conducted by Varvani and coworkers [18]. Local strain data were measured in the vicinity of the notch roots through the use of strain gauges. Strain gauges were mounted to make an approximate distance of 0.5 mm from the grid circuit edge of the strain gauge to the notch root [19]. Asymmetric cyclic tests were conducted on rectangular specimens with dimensions of 100 × 50 × 3 mm. Specimens with different central notch diameters between 9 mm and 15 mm were cyclically tested with a Zwick/Roell HB 100 servo-hydraulic machine. Figure 1 illustrates a drawing of the specimen with a notch diameter of 15 mm. Experiments were conducted under stress-controlled conditions with a stress ratio of R = 0, an asymmetric loading frequency of 0.5 Hz, and at room temperature. Details of ratcheting tests, including the notch diameter, *D*, stress concentration factor, *K_t_*, and nominal stress level $S_{m}\pm S_{a}$ applied on notched specimens, are listed in Table 1. Figure 2 presents the measured ratcheting strains at the notch root of 1045 steel specimens undergoing asymmetric loading cycles for different notch diameters and various stress levels.

Figure 3 compares the measured stress-strain hysteresis loops and local maximum strain data at the notch root of a typical 1045 steel specimen with the loops and local strains generated through analysis. Different coefficients *C*, $\gamma_{1}$, and $\gamma_{2}$ in Figure 3a–c were implemented through several trials to achieve a consistency condition. Figure 3c presents a set of coefficients *C* = 50,000 MPa, $\gamma_{1}$= 350, and $\gamma_{2}=10$ representing a close agreement between the measured and predicted loops, while different values of coefficients in Figure 3a,b resulted in a noticeable difference within the measured loop. Figure 3d plots measured and predicted maximum strain values at the notch root versus loading cycles. The coefficient $\gamma_{2}=10$ resulted in a great agreement between the measured ratcheting data and the predicted curve. The predicted curve position below and above the experimental data for coefficients were $\gamma_{2}>10$ and $\gamma_{2}<10$, respectively.

The measured hysteresis loops in Figure 3d present stress relaxation as the number of stress cycles progressed. A drop in the width of loops with asymmetric loading cycles in this figure verifies the cyclic hardening phenomenon at the notch root of 1045 steel specimen.

## 4. Simulation of Local Ratcheting Strain through Finite Element Analysis

The finite element software ABAQUS version 6.13 [23] was used to simulate the local ratcheting response of steel specimens. Figure 4 shows the meshed specimen undergoing an axial load and its constraints surrounding the circular notch with quadratic elements. Elements were extended in size from the notch root to a distance of 1 mm over the *X*-direction with a mesh size increment of 0.15 mm. The smaller elements were taken at the vicinity of the notch root to achieve a realistic strain/stress comparable with the Neuber, Glinka, and H-S models. The gradual increase in the element size from the notch root enabled them to achieve a better assessment of the strain distribution throughout the modeling process. Translational and rotational axes of the lower end surface of the meshed specimen were constrained (along the *X*- and *Z*-axes) through adapted fixed supports, and the specimen was allowed to take the load along the *Y*-axis. The upper-end surface of the specimen was fixed. The axial load cycles were applied to the lower end of the specimen under a stress-controlled condition with a testing frequency of 0.5 Hz.

The total number of quadratic elements of type C3D8R for the samples with notch diameters of 9 mm and 15 mm were taken, respectively, at 24,306 and 23,748. The former consisted of 29,952 nodes, and the latter possessed 29,670 nodes, respectively. Elements were featured with twenty-four degrees of freedom and with three degrees of freedom per node (eight nodes for each quadratic element). The smallest size of 0.15 mm at the notch root resulted in a consistent convergence as the FE program was run at different applied stress levels and notch sizes. Convergence was consistently achieved for element sizes ranging between 0.15 mm and 0.40 mm, as the program was run for samples during the first hundred loading cycles. For this element, the range size ratcheting at the vicinity of the notch root stayed nearly constant, as presented in Figure 5. In this figure, as elements increased in size beyond 0.40 mm, local ratcheting resulted in decay at different stress levels.

The simulation of ratcheting at the notch root was conducted on the basis of the elastic-plastic materials kinematic hardening model of Chaboche [24]. Based on Chaboche’s non-linear model, the yield surface was translated in the deviatoric stress space as the materials were deformed beyond the elastic limit. The yield surface translation was described based on Chaboche’s postulation as backstress increments were integrated through:(39)$$d\bar{\alpha }=\sum_{i=1}^{3}d{\bar{\alpha }}_{i},\;\;d{\bar{\alpha }}_{i}=\frac{2}{3}C_{i}d{\bar{\varepsilon }}_{p}-\gamma_{i}{}^{\prime }{\bar{\alpha }}_{i}dp$$

Components of backstress $\alpha_{i}$ during the unloading and reloading paths were defined as [24]:(40a)$$\alpha_{i}=\frac{2C_{i}}{3\gamma_{i}{}^{\prime }}+\left(\alpha_{i0}-\frac{2C_{i}}{3\gamma_{i}{}^{\prime }}\right)\exp\left[-\gamma_{i}{}^{\prime }\left(\varepsilon_{p}-\varepsilon_{p0}\right)\right]\;\;d\varepsilon_{p}\geq 0$$
(40b)$$\alpha_{i}=-\frac{2C_{i}}{3\gamma_{i}{}^{\prime }}+\left(\alpha_{i0}+\frac{2C_{i}}{3\gamma_{i}{}^{\prime }}\right)\exp\left[\gamma_{i}{}^{\prime }\left(\varepsilon_{p}-\varepsilon_{p0}\right)\right]\;\;d\varepsilon_{p}<0$$
where $\varepsilon_{p0}$ represents the initial plastic strain and $\alpha_{i0}$ corresponds to the initial backstress. Coefficients $C_{1}$, $C_{2}$, $C_{3}$ and $\gamma \prime_{1}$, $\gamma \prime_{2}$, $\gamma \prime_{3}$ are Chaboche’s materials constants. These coefficients for the 1045 steel alloy were determined from a stress-strain hysteresis loop that was generated based on a strain-controlled test of ±0.8%. Chaboche parameters were obtained by simulating the half, or the lower half of the stabilized hysteresis curve, from the strain-controlled test [24]. The parameter $C_{1}$ was obtained from the slope of the initial part of the stabilized hysteresis curve with a high plastic modulus at the yield point and the parameter $C_{3}\;$was determined from the linear part of the stabilized hysteresis curve with a high strain range. The coefficient $\gamma \prime_{1}$ should be large enough to stabilize the first hardening parameter of Chaboche’s rule. Figure 6 presents an experimentally obtained stress-strain hysteresis loop for the 1045 steel alloy. This figure presents three sets of coefficients $C_{1-3}$ and $\gamma \prime_{1-3}$ and their corresponding loops simulated through FE analysis. These coefficients are chosen to achieve a close agreement between the experimental and simulated hysteresis loops. Figure 6c shows a close agreement of experimental and simulated loops for the 1045 steel alloy for coefficients $C_{1-3}$ = 75,000, 40,000, and 2500 MPa and $\gamma \prime_{1-3}$ = 2200, 215, and 1. Figure 6a,b shows deviations and changes in the simulated loops as different sets of $C_{1-3}$ and $\gamma \prime_{1-3}$ were taken.

## 5. Results and Discussion

The local ratcheting and stress relaxation at the notch root of steel specimens were evaluated through the A-V hardening framework. Local stress and strain components were coupled with the framework through the use of different model choices of Neuber, Glinka, and H-S. Ratcheting at the notch roots was also simulated by FE analysis where Chaboche’s hardening rule was employed.

### 5.1. Local Ratcheting Prediction through the A-V Hardening Rule

#### 5.1.1. Estimation of Local Strain/Stress at Notch Roots through Different Models

To better estimate the local stress and strain terms at the notch root, different choice models of Neuber, Glinka, and H-S were examined. Steel specimens with notch diameters of 9 and 15 mm were tested under nominal stress levels of 155±155 MPa and 203±203 MPa loading conditions. These tests enabled us to evaluate the stress and strain components at the notch root and compare the employed models for their strain energies at applied nominal stresses and at different notch diameters. Figure 7 presents the state of nominal and local stress and strain components for these models at different applied stress levels and notch sizes. In this figure, the pseudo-elastic lines were obtained by applying the constraints of each model, and stress–strain curves were developed based on the Ramberg-Osgood equation. In Figure 7a, the H-S model presented a noticeable increase in the stress and strain components on a specimen with a notch size of 9 mm as the stress level changed from 155±155 MPa to 203±203 MPa. At the given nominal stress level of 155±155 MPa in Figure 7b, the H-S model depicted smaller progress in the local stress and strain components as the notch size of the specimens changed from 9 to 15 mm. A change in the nominal stress level and notch diameter directly affected the extent of stress–strain area based on Glinka and Neuber models. In Figure 7c,d, at higher applied stress levels of 203±203 MPa, the Glinka model resulted in a greater area underneath the stress—strain curve while, for the constant stress level, the specimen with a notch size of 15 mm caused a smaller increase in the local stress/strain data on the curve. Neuber’s rule, however, involved a greater amount of energy from the product of stress and strain obtained from the rectangular area in Figure 7e,f. In the Neuber and Glinka rules, the total strain energy density at the notch root was taken as equal to the total pseudo strain energy density assuming that the specimen did not exceed the elastic domain even beyond its yield point.

#### 5.1.2. Backstress Evolution during Loading

Over the loading paths, backstress evolution was controlled through the A-V kinematic hardening model. Backstress $\bar{\alpha }$ and the internal variable $\bar{b}$ controlled the plastic strain increment $d\varepsilon_{p}$ and its accumulation during asymmetric loading cycles. The magnitude of backstress $\bar{\alpha }$ gradually stabilized over loading cycles in a nonlinear form through the term ($\bar{\alpha }$− 𝛿$\bar{b}$) in the dynamic recovery of the A-V model. This term in the A-V model was analogous to the integration of backstress increments $d\bar{\alpha }=\sum_{i=1}^{3}d{\bar{\alpha }}_{i}$, as proposed earlier by Chaboche [24]. The plastic strain accumulation was attributed to the cross-slip, and as the stress cycles proceeded, the accumulation of dislocations and their interactions led to a decrease in the ratcheting strain rate [26]. Figure 8 shows the evolution of the backstress term ($\bar{\alpha }$− 𝛿$\bar{b}$) over the first thirty loading cycles on a typical 1045 steel specimen through a choice of different models. The decay in the backstress term during stress cycles was more pronounced as the Neuber and Glinka rules were coupled with the A-V model compared to that of the H-S model. This figure shows a sudden drop in term ($\bar{\alpha }$− 𝛿$\bar{b}$) over the first few cycles. Following the initial loading cycles, a steady state was achieved. The smaller difference in terms ($\bar{\alpha }$− 𝛿$\bar{b}$) between the Neuber and Glinka rules was associated in relation to the nominal and local stresses through Equation (13). A larger product of the local stress and strain components in Figure 7e,f resulted in a larger nominal stress component and a small drop in term ($\bar{\alpha }$− 𝛿$\bar{b}$) over the loading cycles. The lowest backstress term in Figure 8 was attributed to the equivalent stress components, as defined in the H-S model. In the presence of the H-S model, the backstress term achieved its stability after the first seven cycles, while backstress curves that were generated through the use of Neuber and Glinka models required an even smaller number of cycles to gain a steady condition.

### 5.2. Predicted and Simulated Local Ratcheting Curves

The coupled kinematic hardening framework was employed to assess the ratcheting response of notched 1045 steel specimens undergoing asymmetric stress cycles. The predicted and experimental ratcheting results at various stress levels and notch sizes were plotted in Figure 9. The predicted ratcheting curves in this figure show a consistent response compared with the experimental data. Predicted local ratcheting over the first few cycles showed an abrupt increase, and shortly after, as the number of cycles increased, the ratcheting rate dropped, and the slope of the ratcheting progress stayed nearly constant. Considering Glinka, Neuber, and H-S models, the choice of the H-S model resulted in lower ratcheting curves, and those curves were predicted on the basis of the hardening framework coupled with the Glinka model, which possessed the highest ratcheting values. The H-S model has, however, shown closer agreements with the experimental data at different stress levels and notch sizes over the first 20–40 cycles. Ratcheting curves predicted by the coupled framework of the A-V hardening rule and Neuber model closely agreed with the measured values of ratcheting strains over the loading cycles. Lower local stress and strain at the notch root were calculated based on the H-S model and suppressed the predicted ratcheting curves by the hardening framework. Higher strain energy was achieved through the Neuber and Glinka models, which increased the predicted local ratcheting as local stress and strain values at the notch root increased. Figure 9 shows the simulated ratcheting results through the use of FE analysis for 1045 steel specimens with different notch sizes undergoing different stress levels, which fell below the experimental and predicted ratcheting curves. Simulated curves were affected by the FE element size taken at different distances from the notch root to achieve a consistent convergence. The simulated curves correspond to lower local ratcheting values with a noticeable difference between the measured ratcheting data and those of the predicted curves.

The test samples with the same diameter consistently showed that an increase in the applied stress led to a higher ratcheting strain. At the same applied stress level, an increase in the diameter of the specimen from D = 9 mm to 15 mm decreased the ratcheting strain noticeably. While the ratcheting strain was promoted during asymmetric stress cycles in Figure 9, the cyclic stress levels gradually dropped to lower levels revealing stress relaxation at the notch root of the steel specimens. Stress relaxation over the stress cycles at the notch root of specimens was found at slightly different rates as the A-V hardening model was coupled with different models. Hysteresis loops of a typical 1045 steel specimen with a notch diameter of D = 9 mm were tested at 203±203MPa and showed that the H-S model resulted in the widening of loops while the Glinka model lowered the plastic strain range in the progressing loops. Loops generated by the hardening rule coupled with the Neuber model possessed an intermediate width. These hysteresis loops are presented in Figure 10. It is intended to keep the stress and strain axes in this figure within the same scale for a better comparison of these models and their generated loops. The peaks of progressing loops were connected as the number of cycles proceeded in Figure 11, resulting in a decreasing trend with nearly the same pace for all models. The horizontal axis in these figures was normalized with the strain at the 100th cycle to be able to present stress relaxation when three different models were compared. The ratcheting progress was controlled through the hardening framework at constant stress cycles, while stress relaxation at the notch root was controlled at a constant local strain leading to progressive loops experiencing a drop in the stress magnitude. Predicted curves in Figure 11 show relatively lower values of stresses and noticeable rates of relaxation compared with the simulated curve through the FE analysis. In this figure, the simulated stress relation for cycles beyond the first 20 cycles is presented for a consistent comparison with the predicted curves.

### 5.3. Discussion

Ratcheting strain progress with asymmetric loading cycles took place over the transient and steady stages. Over stage II, ratcheting was found to be associated with the coefficient δ in the dynamic recovery of the A-V hardening rule. This coefficient was defined to be related to the backstress evolution and materials constants, *C*, $\gamma_{1}$, and *m* through ${\left(\bar{\alpha }/k\right)}^{m}$. The local ratcheting rates (slopes) were predicted based on the coupled framework with different models and were found to be relatively smaller than that of the experimental data for stage II.

Various cyclic tests were conducted to examine the local ratcheting response of 1045 steel specimens at the vicinity of notch roots. While the present authors studied a number of influential parameters on the local ratcheting in the presence of the notch in steel plates, including stress levels and notch sizes, the open literature lacks a pertinent volume of experimental and theoretical research on local ratcheting phenomenon. More investigation is required to fully address the local ratcheting and stress relaxation at the notch root. Challenges in research include the complexity of cyclic tests to examine/detect the notch root plasticity and accurately measure local strains, as well as the lack of hardening frameworks/theories to sufficiently address the plastic flow in the vicinity of notch roots. The authors believe that more experimental investigation and analysis are required to fully understand notch root plasticity and its progress during asymmetric loading cycles. To measure progressive plastic deformation at the notch root and at various distances from the notch root, the use of strain gauges along longitudinal and lateral directions are inevitable. The author’s further plan is to control the local ratcheting progress through technical/mechanical processes, including cold-pressing the notch root and localized heat-treatment processes. While such processes are expected to improve mechanical properties at the notch edges, they will also noticeably lessen the ratcheting magnitude and rate. The coupled hardening model, along with the Neuber and Glinka rules, will be employed to assess the ratcheting of cold-worked notches. Analytical and numerical approaches will be used to evaluate the choice of hardening rules and to encounter more variables such as time dependency, stress rate, testing frequency, and temperature in the ratcheting assessment program.

## 6. Conclusions

Local ratcheting was evaluated at the notch root of 1045 steel specimens by means of the A-V hardening rule coupled with the Neuber, Glinka, and H-S models. The evolution of backstress was governed by the coupled hardening framework. The local ratcheting rate and magnitude and stress relaxation at the notch root of steel specimens was assessed at different stress levels and notch sizes through various coupled models and FE analysis. The hardening rule algorithm was developed to assess local ratcheting coupled with different model choices to assess local strain and stress at the notch root. Predicted ratcheting curves through the coupled hardening framework with the Glinka model shifted above the measured ratcheting data, and those evaluated by means of H-S fell below experimental data. The Neuber model, however, closely agreed with the experimental ratcheting data at different stress levels and specimen notch sizes. Local ratcheting simulated through FE analysis fell below the experimental data and predicted curves. The choice of Neuber, Glinka, and H-S models in assessing local strain components and their terms/constants was found to affect the rate and magnitude of predicted ratcheting and stress relaxation by means of the coupled hardening framework.

## Figures and Tables

**Figure 1 materials-16-02153-f001:**
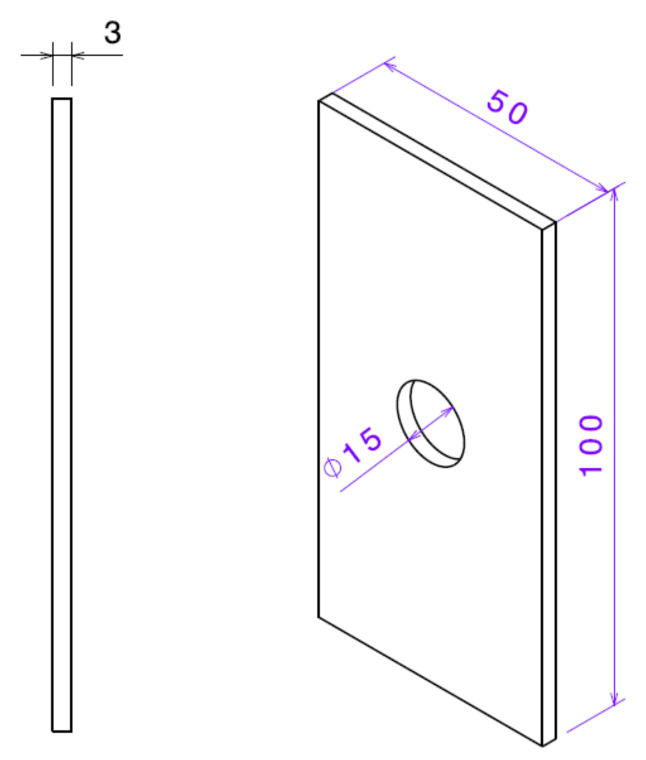
Drawing of the specimen with 15 mm notch diameter (dimensions are in mm).

**Figure 2 materials-16-02153-f002:**
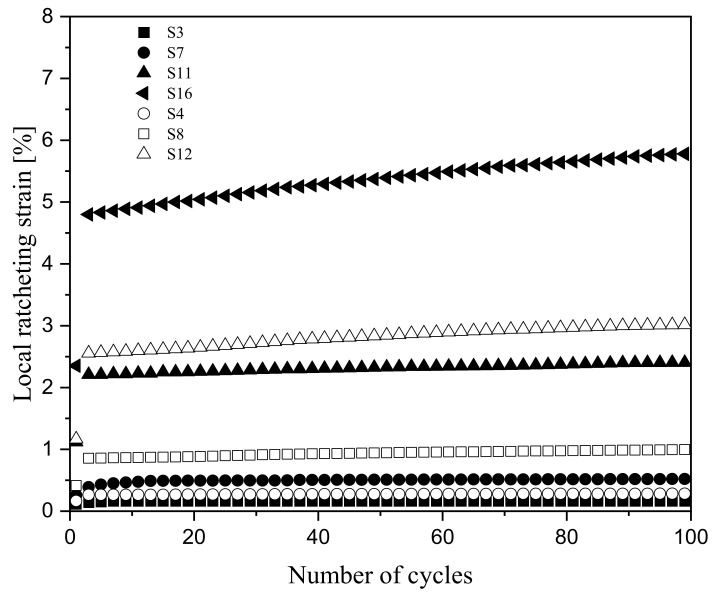
Ratcheting data collected at notch roots of 1045 steel specimens with different notch diameters and stress levels [18].

**Figure 3 materials-16-02153-f003:**
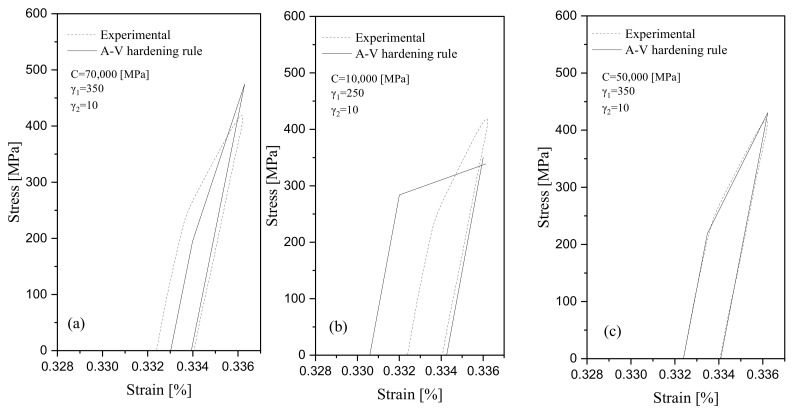
Determination of coefficients C, $\gamma_{1}$, and $\gamma_{2}$ to achieve consistency condition based on the A-V model. (**a**–**c**) several trials of coefficients *C*, $\gamma_{1}$, and $\gamma_{2}$ to achieve a consistency condition, and (**d**) measured and predicted maximum strain values at the notch root versus loading cycles.

**Figure 4 materials-16-02153-f004:**
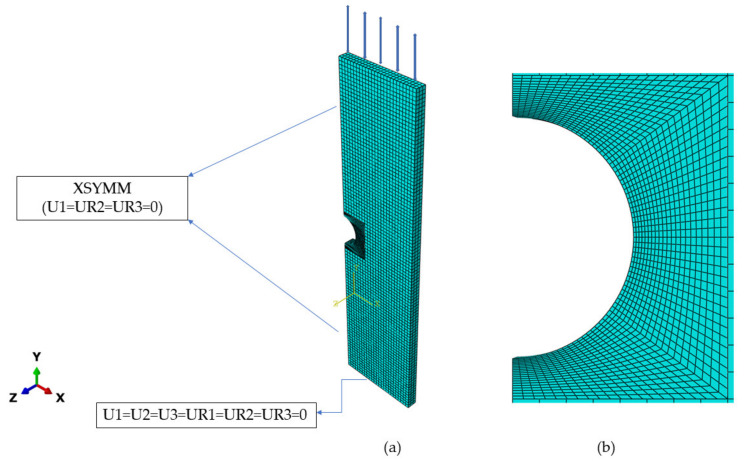
(**a**) Meshed specimen loaded axially along *Y*-axis and boundary conditions, (**b**) Meshing at the vicinity of notch root through quadratic elements.

**Figure 5 materials-16-02153-f005:**
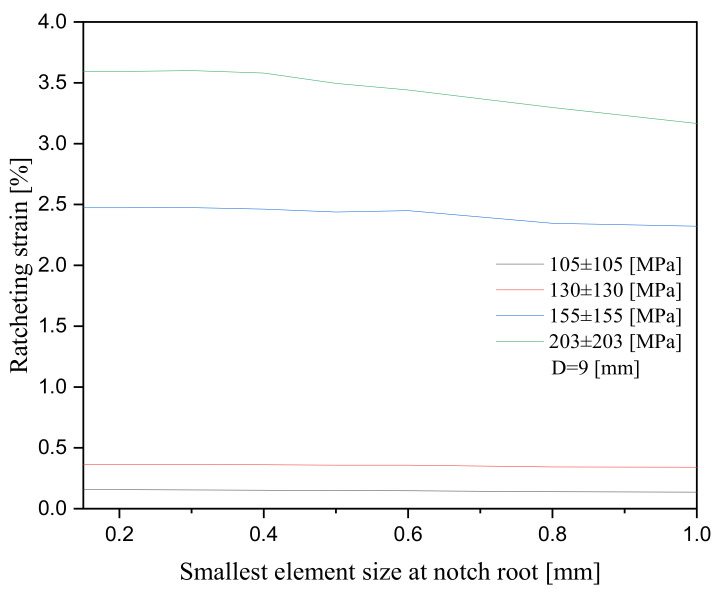
Convergence of ratcheting strain at notch root of the 1045 steel specimen (D = 9 mm) versus the quadratic mesh size at different stress levels for a given hysteresis loop.

**Figure 6 materials-16-02153-f006:**
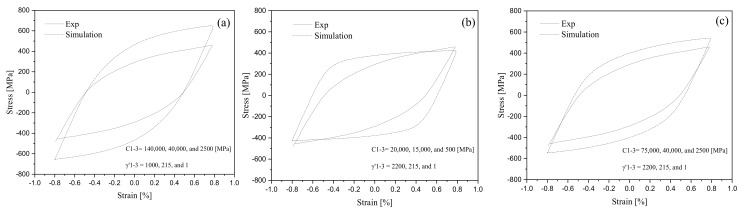
Coefficients $C_{1-3}$ and ${\gamma \prime }_{1-3}$ achieved a close agreement between the stress-strain hysteresis loop obtained from a test conducted under the strain-controlled condition and the one simulated through FE analysis [25]. (**a**–**c**) the strain-based hysteresis loops for different sets of $C_{1-3}$ and $\gamma \prime_{1-3}$.

**Figure 7 materials-16-02153-f007:**
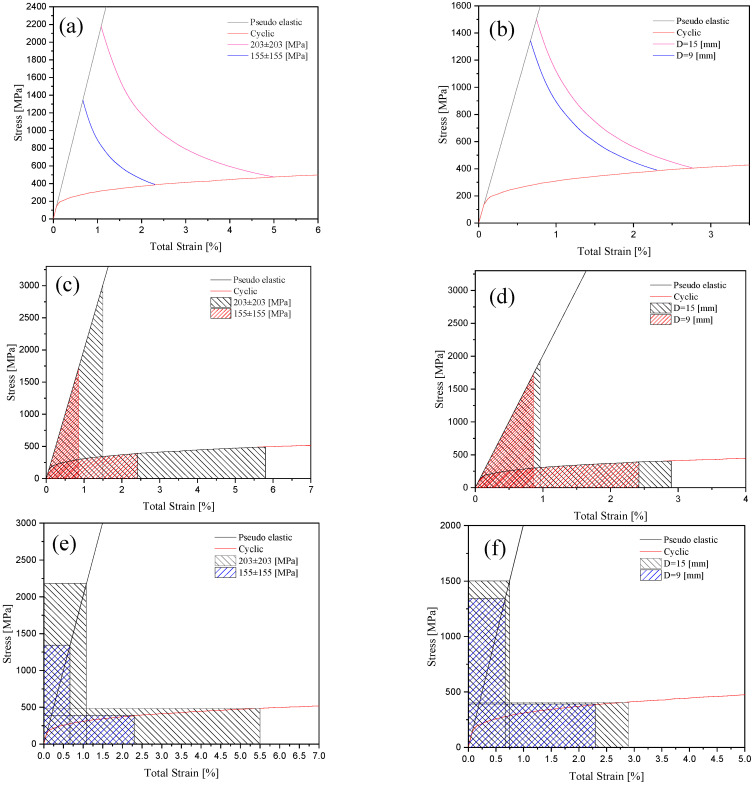
(**a**,**c**,**e**) Cyclic and pseudo–elastic curves for steel specimens with D = 9 mm notch diameter subjected to different loading values using H-S, Glinka and Neuber models. (**b**,**d**,**f**) cyclic and pseudo–elastic curves for a steel specimen undergoing 155±155 MPa for notched specimens with D = 9 and D = 15 mm notch diameters using the H-S, Glinka, and Neuber models.

**Figure 8 materials-16-02153-f008:**
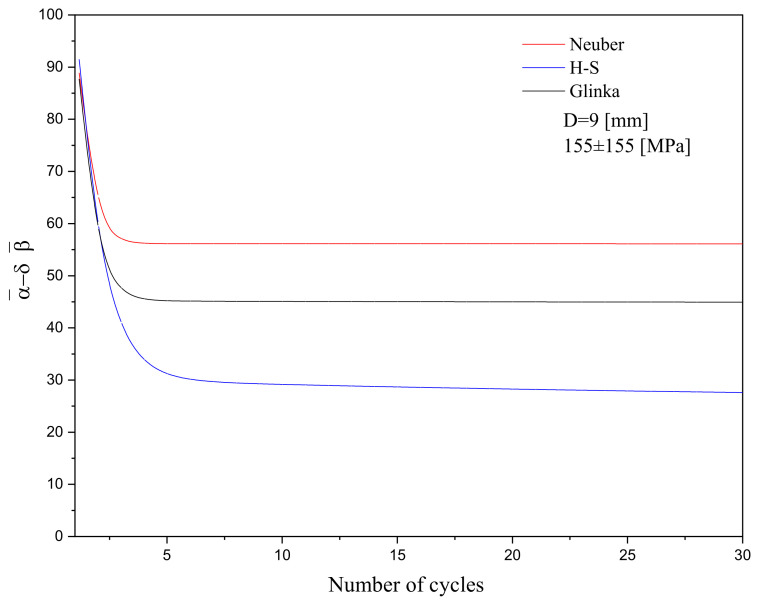
Backstress evolution over the first 30 loading cycles based on the Neuber, H-S, and Glinka rule coupled with the A-V kinematic hardening rule for a steel specimen and an 8 mm diameter under 155±155 MPa.

**Figure 9 materials-16-02153-f009:**
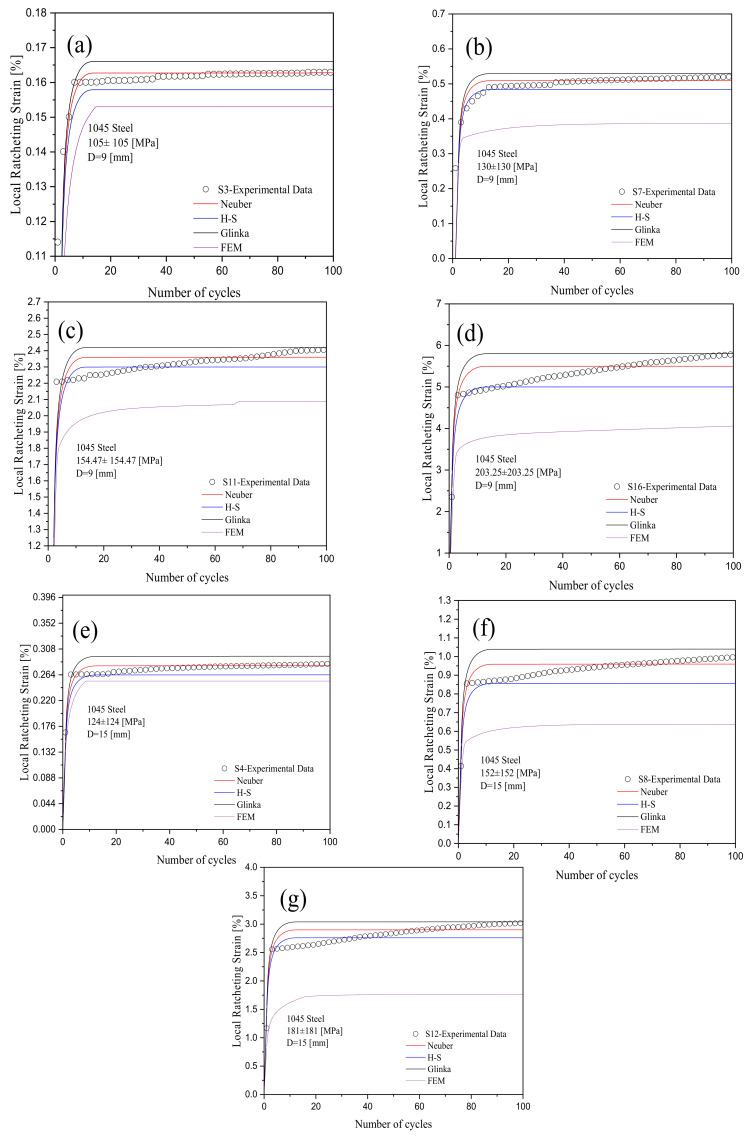
Predicted ratcheting curves using three different models coupled with the A-V hardening rule versus experimental ratcheting data for 1045 steel notched specimens undergoing various stress levels of (**a**) 105±105 MPa with D = 9 mm, (**b**) 130±130 MPa with D = 9 mm, (**c**) 155±155 MPa with D = 9 mm, (**d**) 203±203 MPa with D = 9 mm, (**e**) 124±124 MPa with D = 15 mm, (**f**) 152±152 MPa with D = 15 mm, (**g**) 181±181 MPa with D = 15 mm.

**Figure 10 materials-16-02153-f010:**
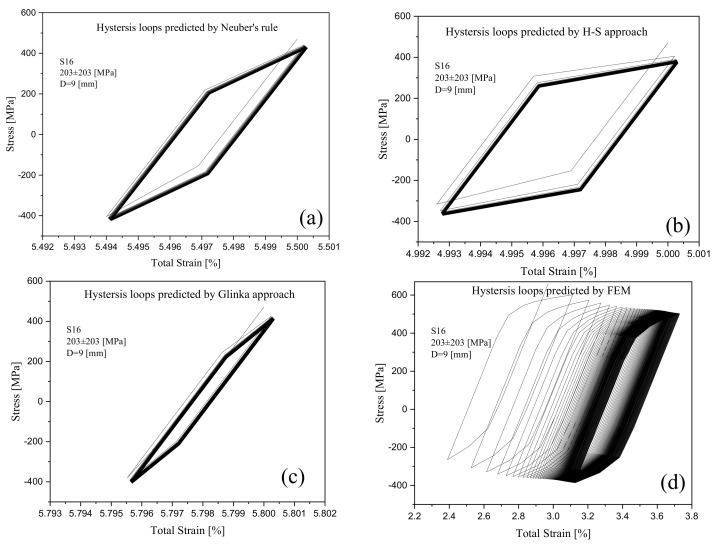
(**a**–**c**) Predicted stress-strain hysteresis loops based on Neuber’s rule, H-S rule, and Glinka’s approach coupled with the A-V model, respectively, and (**d**) simulated stress-strain hysteresis loops by FE analysis on the basis of Chaboche’s model.

**Figure 11 materials-16-02153-f011:**
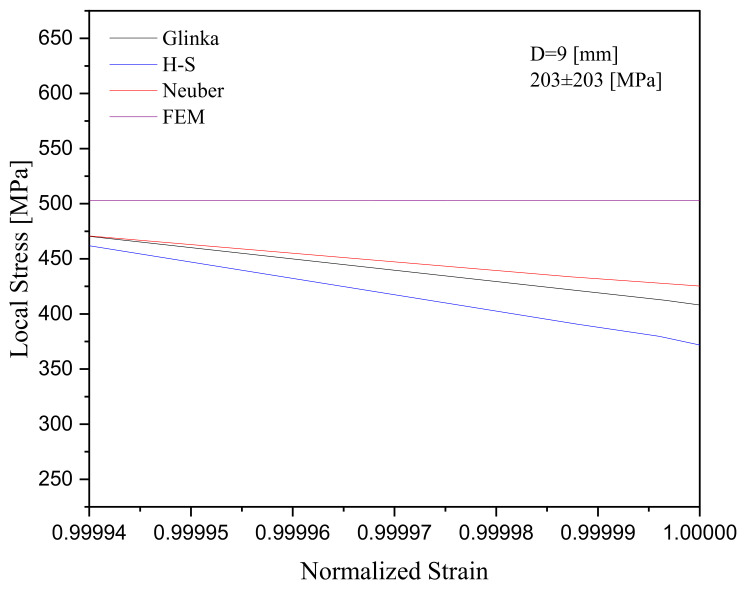
Stress relaxation for a steel specimen with a 9 mm notch diameter subjected to 203 ± 203 MPa, using three different rules coupled with the A-V kinematic hardening rule and through FE analysis.

**Table 1 materials-16-02153-t001:** Loading condition for 1045 steel specimens with different notch diameters [18].

Test Specimen	Notch Diameter (D) (mm)	$K_{t}$	$S_{m}\pm S_{a}(\mathrm{MPa})$
S3	9	2.53	105±105
S7	9	2.53	130±130
S11	9	2.53	155±155
S16	9	2.53	203±203
S4	15	2.36	124±124
S8	15	2.36	152±152
S12	15	2.36	181±181

## Data Availability

Experimental data can be available upon request of researchers through communication with corresponding author.

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
