# Peer review of "Accumulation of Plastic Strain at Notch Root of Steel Specimens Undergoing Asymmetric Fatigue Cycles: Analysis and Simulation"

_materials, 2023, doi:10.3390/ma16062153_

Round 1

Reviewer 1 Report

Comments and Suggestions for Authors are given in the attached doc file.

Author Response

General remarks by the Reviewer 1:

1. It is always interesting to read papers involving experiments, as they can give a lot of interesting information. Still, this manuscript is characterized by inconsistencies, mainly with variables and their naming.

1. Authors's response:

Authors thank the reviewer for the constructive comments made by the reviewer. Terms/ variables used in the manuscript have been clarified in details as suggested by the reviewer.

2. Due to the style consistency, all variables in the text of the manuscript should be written in italics (and most of them actually are). 

2. Authors's response:

Variables in the text of the manuscript were written in italics.

3. The letter “d” stands in Eq. (1) for the derivative (probably, because it is actually not explained), but in Chapter 3, it represents the notch diameter (and at the end of Chapter 5 again, something undefined else).

3. Authors's response:

To differentiate between derivative d and notch diameter d, the notch diameter is denoted as D in the manuscript.

4. It is also hard to distinguish a single variable denoted with two symbols (for example, “ ”) from a product of two variables which visually looks almost the same (for example, “ ”). The equations are challenging to understand and verify, as all the variables that appear are not explained (or their descriptions are given somewhere later in the article - for some even twice)

4. Authors's response:

To address all variables in the manuscript, a nomenclature is added after the Abstract section as suggested by the reviewer.

5. The phrase “et al.” is in the context of an academic paper more commonly used than “and coworkers”. Further, when referring to a reference in the manuscript, the first author should be quoted. Several double spaces pop out of the manuscript.

5. Authors's response:

Authors naming and references were modified in the manuscript, as suggested by the reviewer.

Chapter 2:

6. Further, the major problem is that the reader has to find out by himself whether a variable presents a scalar or a tensor (matrix). Although I do admit that mathematically a scalar is indeed a zero-rank tensor, I also further think that describing scalars as tensors in structural mechanics papers does not contribute to the overall readability of the manuscript. Furthermore, the Author cannot expect that all potential readers will be experts in the field and, consequently, know all these details. Eq. (1) introduces a sum of two variables: either scalars or tensors (matrices). Although some indications suggest that all three variables are scalars (the steel samples considered are exposed to uniaxial loading, and each variable is generally described as a strain increment and not as a strain increment tensor), the first term of Eq. (2) (a second rank tensor divided by a number and a scalar) reveals that they are actually (second rank) tensors.

6. Authors's response: 

Equations (1) and (2) and their variables were described and listed in the nomenclature, as suggested by the reviewer.

7. However, variables E, G and J are not explained. Although these variables are commonly used in structural mechanics (except J), the Authors should not rely on this and should nevertheless be explained in the manuscript as soon as they appear. For example, the meaning of the variable E, which appears in Eq. (2) on page 2, is not given until Line 129 on page 4.

7. Authors's response: 

Variables and terms of equations were defined in the manuscript and listed in the nomenclature as suggested by the reviewer.

8. In the bracket of Eq. (2) is the dot product of two tensors: the increment of the stress tensor by the unit tensor. Assuming that the unit tensor is equal to the identity matrix (even our mathematician cannot offer another possibility), the product in the brackets further directly equals the increment of the stress tensor (i.e. there is no need for this product). However, if the unit tensor somehow does not equal the identity matrix, this tensor should be either adequately described or (even better) presented.

8. Authors's response:

Terms of Equations (1) and (2) were defined in the manuscript and listed in the nomenclature.

9. Furthermore, the expression in the parentheses (the dot product of a matrix with a vector) of Eq. (3) produces a vector further multiplied by the normal vector. But, the first term (a tensor) at the right-hand side of Eq. (2), which is introduced in Eq. (1), shows that also the result of Eq. (3) should be a tensor.

9. Authors's response:

As suggested by the reviewer, Equation (3) was defined as a tensor in the manuscript.

10. The expression in both identical parentheses of Eq. (4) is a difference between the increment of the deviatoric tensor and the backstress tensor and is, therefore, also a tensor. Consequently, the right side of Eq.(4) represents a product of two tensors from which a scalar is being subtracted (assuming that σy represents the yield limit, which is usually denoted as σY to distinguish it from the normal stress in the y direction).

10. Authors's response:

Equation (4) represents the yield surface using von-Mises yield criterion [1-4]. Yield surfaces can translate but cannot rotate and their shapes do not change during loading or unloading process. Yield surfaces must satisfy the consistency condition in order to follow the stress points in the deviatoric stress space. In equation (4)  defines the size of the yield surface to translate and yield surface maintains its shape.

[1] Khan AS, Huang S. Continuum theory of plasticity. John Wiley and Sons 1995.

[2] Ahmadzadeh GR, Varvani-Farahani A. Ratcheting assessment of materials based on the modified Armstrong–Frederick hardening rule at various uniaxial stress levels. Fatigue Fract Eng Mater Struct 2013;36(12):1232-1245.

[3] Lee CH, Van Do VN, Chang KH. Analysis of uniaxial ratcheting behavior and cyclic mean stress relaxation of a duplex stainless steel. Int J Plasticity 2014;62:17-33.

[4] Kolasangiani K, Farhangdoost K, Shariati M, Varvani-Farahani A. Ratcheting progress at notch root of 1045 steel samples over asymmetric loading cycles: experiments and analyses. Fatigue Fract Eng Mater Struct 2018;41(9):1870-1883.

11.In the parentheses in Eq. (5-a), a product (= a scalar) of material-dependent coefficient d (a scalar since it is being defined as a constant) by the internal variable  (a scalar) is being subtracted from backstress tensor  (according to Line 100). However, in Line 112, variable  is described as (a single value or a scalar) backstress.

11. Authors's response:

Internal variable, scalar term, and backstress were clarified in the manuscript as suggested by the reviewer. 

12. The internal variable  is being mentioned in several places in the manuscript, but no details regarding its evaluation or range values are provided.

12. Authors's response:

The internal variable was defined in the manuscript and a reference for further details was added in the manuscript.

13.Eq.(6) defines the calculation of variable dp (presumably a scalar) as a root of the dot product of two tensors (which is mathematically also a tensor).

13. Authors's response:

Terms of Equations (6) were checked as suggested by the reviewer.

14. The material-dependent coefficient d (a scalar) is defined through a ratio of a tensor and a scalar value, which is again a tensor.

14. Authors's response:

Coefficient d was checked as suggested by the reviewer.

15. The parameters S, s  and e appearing in Eqs. (8) and (9) are not adequately described – nominal and local stresses are mentioned after Eq.(12), while the nominal strain is not explained anywhere.

15. Authors's response:

Parameters S, s  and e were corrected as suggested by the reviewer.

16. While the subscripts A, B and C from Eq. (13) and (14) are almost correctly explained (the Authors should namely make a simple comment to help the reader to distinguish the maximum load in point A from the maximum load in point C), the meaning of subscript L remains unknown (it can be either longitudinal or local – in the case of the latter, this subscript is probably even redundant). This subscript also does not appear in similar equations obtained by Glinka’s rule (Eqs. (27)-(28)).

16. Authors's response:

Subscript “L” denoted local components. To keep consistency of terms, nomenclature lists  and   as local stress and strain terms at notch root. Subscript “L” was removed from the text.

17. The problem with Eqs. (13)-(14) (and (27)-(28) & (37)-(38) as well) is that the parameter that governs the utilisation of either equation is , which is, according to Eqs. (1) and (2), a tensor (as already mentioned above).

17. Authors's response:

These equations were clarified as materials testing was conducted under uniaxial conditions.

18. The Authors claim that Eq. (15) relates the uniaxial nominal strain and stress. However, in my opinion, the way this equation is being written, it actually relates the uniaxial nominal strain and stress differences (increments or decrements) or ranges.

18. Authors's response:

Ramberg-Osgood equation was employed. To further clarify it, “range” of stress and strain components were used in the manuscript.

19. Since in Eqs. (18)-(20) the a coefficient is being presented for the first time (these three coefficients are scalars, while the similar symbol  represents a tensor) it deserves some explanation.

19. Authors's response:

It is explained in the text that “for components subjected to uniaxial loading, local backstress for loading half-cycle can be related to local stress at turning points”. Therefore, the backstress component in the loading direction is considered. The word “component” is added to further clarify it in the manuscript.

20. Since the Authors use expressions “strain energy density” as well as “strain energy per unit volume”, they should explain the difference between them.

20. Authors's response:

Strain energy is defined as the energy stored in a body due to deformation. The strain energy per unit volume is known as strain energy density and the area under the stress-strain curve towards the point of deformation. When the applied force is released, the whole system returns to its original shape. In the manuscript “strain energy density” and “strain energy per unit volume” are identical terms.

21. The Authors should give some details of the s(e) function appearing in the integral in Eq. (23).

21. Authors's response:

Details of s(e) was given in manuscript as suggested by the reviewer.

22. The local stress and strain at the notch root are in Line 169 now clearly named uniaxial quantities. I think this information should appear already in Line 129.

22. Authors's response:

This definition now is provided in line 129 of the manuscript as suggested by the reviewer. 

23. The Authors should clearly note that Eq. (31) is valid only in the absence of any shear stresses. Further, I think that components sex, sey and sez should be named more precisely as elastic (normal) components since the subscript e might mean either elastic or equivalent.

23.Authors's response:

The editorial corrections were made in the manuscript as suggested by the reviewer.

24. The equations that belong to the H-S approach (method) provide the information on how to obtain the theoretical elastic equivalent stress at notch root seq (Eq. (31)), and the equivalent stress concentration (probably factor) Ktq (Eq. (34)), but no information how to obtain sq, which appears in both equations for the determination of eq, Eqs. (29) as well as (36). Further, Eq. (29) contains a function F(), which is neither presented nor properly discussed. The only information provided is that the function (values?) falls between the given range. However, this might not be a severe handicap if the H-S method and the H-S approach are the same technique, which consequently means that the purpose of two equations for the determination of the same equivalent strain is intriguing. A similar situation also appears with Eqs. (34) and (35), where the purpose of providing two equations for the determination of the same equivalent stress concentration (factor?) is unknown.

24. Authors's response:

Equation (29) is the general and initial form of the H-S model, equations 37 and 38 have been developed as applied stress and strain terms were related to local components through H-S model for unloading and reloading paths. Hoffman-Seeger initially expresses the equivalent strain as a function of F and evaluates the F function for different cases (Elastic Perfectly-Plastic material, strain hardening material). That is, in the end, the general form of equation (29) for strain hardening material becomes equation (37) and (38). The purpose of bringing equation 34 is to first introduce the equivalent stress concentration factor and then relating it to the stress concentration factor through the von-Mises criterion.

25. In the subchapter describing the ratcheting analysis algorithm, the backstress (component)  is being mentioned. However, in two parentheses of this subchapter, it was replaced by  without any explanation of this new variable.

25. Authors's response:

Editorial corrections were made as suggested by the reviewer.

Chapter 3:

26. In most of the manuscript, the Authors use the word “asymmetric” with the word “loading”; therefore, I think this should also be done in Chapter 3.

26. Authors's response:

Editorial modifications were made as suggested by the reviewer.

27. The Authors should explain if the strain gauge width was 1 mm which would allow it to measure the local strain at a distance x=0.5 mm from the notch roots, or just the border of the strain gauge was located at a distance x= 0.5 mm.

27. Authors's response:

The grid circuit's edge of strain gauge was positioned 0.5mm from the notch root. This has been clarified in the manuscript, as suggested by the reviewer.

28. In Table 1, a dimension should be specified for the diameter d values (it is true that this information is given above the table, but I think it should also be repeated in the table).

28. Authors's response:

Editorial changes were made as suggested by the reviewer.

29. Coefficients C and  g1 were in subchapter 2.2, generally introduced as parameters defined from uniaxial stress-strain hysteresis loops. For these coefficients (as well as for g2 – a material-dependent coefficient), some values are presented in the text below Figure 1 (describing Figure 2), but their actual meaning and application are not revealed. These values were simply presented as the values representing a close agreement between measured and predicted loops in figure 2(c), while different values of coefficients in figures 2(a) and 2(b) resulted in a noticeable difference with the measured loop. However, the methodology for selecting their combinations for application should be explained (or were these values simply randomly selected?). Finally, since coefficient C has units, it certainly has an engineering/physical meaning.

29. Authors's response:

These coefficients depend on the mechanical properties of the material and are unique for each material. The method to determine these coefficients is described in the manuscript. Several sets of C and  were evaluated to achieve consistency condition as illustrated in the new figure 3.

30. The Authors should explain the purpose of inserting a figure within Figure 2(d).

30. Authors's response:

Finding coefficients C, , and  to achieve consistency condition enabled us to see two results in the hysteresis loops:

- As the number of stress cycles increases, stress relaxation is visible in the observed hysteresis loops in figure 2(d).

- The cyclic hardening phenomena at the notch root of the 1045 steel specimen is confirmed in this image by a reduction in the width of loops with asymmetric loading cycles.

Chapter 4:

31. The considered steel samples, as well as FE model analyses, require better description. Firstly, the dimensions of the steel sample should be provided. Further, the FE model basic data should comprise the number of finite elements, degrees of freedom applied, and the number of equations solved for each axial load cycle.

31. Authors's response:

The details for FE analysis have been provided in the manuscript as suggested by the reviewer.

32. Further, the Authors quote that convergence was consistently achieved for element sizes ranging between 0.15 mm and 0.40 mm through several runs, but the number of cycles required is not given.

32. Authors's response:

To achieve convergence, samples were subjected to 100 cycles were for a given element size. Ratcheting data of cycles 1, 20, 50, 70 and 100 were evaluated. The manuscript however presents the convergence diagram for the first cycle.

33. The sentence “The gradual increase in element size enabled to achieve a better assessment of strain distribution throughout the modelling process.” needs some explanation.

33. Authors's response:

The smallest element size was 0.15 mm and as the distance from the notch root in x-direction increased, the size of the elements increased. This has been clarified in the manuscript, as suggested by the reviewer.

34. The text below Figure 4 is written in smaller font size (9 instead of 10).

34. Authors's response:

Editorial and font size were corrected as recommended by the reviewer.

35. In the middle part of Eq. (39), a summation symbol is seen, but this summation symbol is not presented in the last part of Eq.(39), although the subscript i remains (or a comma and an additional space should replace perhaps the dot symbol). Further, Eq. (39) contains coefficients  and , but Eqs. (40) define  coefficients.

35. Authors's response:

Editorial corrections were made as suggested by the reviewer.

36. Coefficients g’1, g’2 and g’3 are presented as Chaboche’s material constants, but Eqs. (40) contain (the same?) coefficients without the apostrophe.

36. Authors's response:

Coefficients and symbols were revised as suggested by the reviewer.

37. Further, while for the three (out of six) Chaboche’s material constants, at least some brief information regarding their evaluations (although probably not good enough for any reader to actually implement them) are given (C1, C3 and g’1), any further information regarding the acquiring the rest of them is not provided.

37. Authors's response:

These coefficients for 1045 steel alloy were derived using a stress-strain hysteresis loop produced by a strain-controlled test with a range of . The lower half of the stabilised hysteresis curve from the strain-controlled test is used to simulate Chaboche parameters. More information as well as reference for readers are given after equation (40).

38. Although I certainly do agree that Figure 2c shows a close agreement between measured and predicted loops, I would not dare to say that any of Figures 5 shows a close agreement between experimental and simulated loops (like the Authors claim for Figure 5(c)). However, on the other hand, I also would not describe any of Figures 5 as particularly bad, i.e. Figures 5(a) and 5(b) are not worse or better than Figure 5(c), for which the Authors claim is far better than the others.

38. Authors's response:

Figure 6 (c) has the smallest difference with the experimental values and also maintains the shape of the hysteresis loop. In the third diagram, there are more common points between the simulated loop and the experimental loop at both ends.

Chapter 5:

39. I think that the descriptions in Figures 6 should be synchronised with the caption of these figures. The descriptions in Figures 6 namely compare Cyclic and Pseudo-elastic curves, while the caption quotes Actual and Pseudo-elastic curves.

39. Authors's response:

Editorial corrections were made as suggested by the reviewer.

40. In Line 355, the plastic strain increment is denoted slightly differently as in the rest of the manuscript.

40. Authors's response:

Editorial revisions were made as suggested by the reviewer.

41. In Line 358, the backstress increment is presented as a sum of some terms, similar to Eq. (39). However, the main difference is that in Eq. (39), the upper limit of summation is clearly given, while in Line 358, the upper limit of summation is neither defined nor mentioned.

41. Authors's response:

The upper limit of summation in line 358, M, is the number of backstress increments which is 3 for Chaboche’s rule. It is changed to 3, in line 358.

42. In Figure 7, the Hoffman and Seeger rule (approach) is denoted as Hoffman, while in the rest of the manuscript is denoted as H-S (this remark also applies to Figure 9. (b)).

42. Authors's response:

Editorial corrections were made as suggested by the reviewer.

43. Further, the value on the ordinate must be written as everywhere else in the manuscript, i.e.: (a-db)

43. Authors's response:

Editorial corrections were made as suggested by the reviewer.

44. I think that the sentence “The coupled A-V/H-S framework suppressed the magnitude of plastic deformation and ratcheting over initial loading cycles.” is somehow incomplete since it can be misinterpreted as referring to the measured values of plastic deformation and ratcheting (rather than their numerical simulations obtained obtained by implementing Neubert’s and Glinka’s rule).

44. Authors's response:

This sentence is omitted.

45. Figure 8 – due to relatively small figures, I advise the Authors to use colours instead different dots or dashed lines as these are quite challenging to read (even on the computer screen and not just printed on the paper). The identical colours might have also been used in Figures 7 and 10, although the problem of distinguishing lines is not so evident there.

45. Authors's response:

Editorial corrections were made as suggested by the reviewer.

46. Further, since Figure 8 consists of seven individual figures, there is no need to use different symbols for experimental data among figures.

46. Authors's response:

Corrections were made as suggested by the reviewer.

47. Although it is quite obvious what the Authors wanted to say while describing Figure 8, it is still improper to have a stage II without any mentioning and clearly defining stage I.

47. Authors's response:

A brief definition of stages is added to the manuscript, as suggested by the reviewer.

48. Further, the Authors quote, "The choice of H-S model resulted in the lowest ratcheting curves", which is true only if the three models are considered, as the FEM model (based on Chaboche’s model?) actually produced the lowest curves.

48.  Authors's response:

Editorial revisions were made as suggested by the reviewer.

49. I cannot simply agree that generally (except for figure 8(g)) “The H-S model has however shown closer agreements with experimental data at different stress levels and notch sizes over the first 20-40 cycles.” as these agreements are not good enough to call the H-S model (which is figures entitled just HS) the “winner” (even for the first 20-40 cycles). Further, although I do somehow agree that the Neuber’s model can be called the least worse (and certainly not the “closely agreed” model), I think some numerical analyses should support this conclusion. Finally, the information from the last sentence prior to Figure 8 should be completed by revealing whether the presented FE results are the best or the worst among all obtained.

49. Authors's response:

Simulated results were discussed in section 5.2 along with the predicted ratcheting data. The simulated curves correspond to lower local ratcheting values with a noticeable difference from measured ratcheting data and those of predicted curves.

50. Hysteresis loops of a typical 1045 steel sample with a notch diameter of d=9 mm tested at 203±203MPa are presented in Figure 9. However, from previous analyses, it is evident that the Authors also studied the notch diameter of d=15 mm, as well as different stress levers (this remark is also valid for Figure 10). Therefore, if these results are not presented, they should be at least commented on to realise the impact of these parameters on both the predicted stress-strain hysteresis loops as well as the stress relaxation.

50. Authors's response:

The purpose of this figure was to compare hysteresis loops and relaxation stress using different models. That’s why the second geometry was not presented.

51. In Lines 436-438 the Authors describe the ratcheting over stage II and mention the coefficients d and g1, which, however, do not appear in subchapter 2.2. Further, the last sentence of the first paragraph below Figure 10 seems to describe Figure 8. (a). If this is correct, this figure should be clearly mentioned. Nevertheless, the complete text below Figure 10 sounds more like the Conclusion than the Predicted and simulated local ratcheting curves. Therefore, this part could perhaps become section 5.3 Discussion. Even so, it would be interesting to know why the Authors will continue to use only Neuber's and Glinka's rules and abandon the H-S rule.

51. Authors's response:

The last sentence of the first paragraph below the Figure 10 is deleted and the section is changed to part 5.3 Discussion, as suggested by the reviewer.

Chapter 6:

52. Further, I find the sentence “The predicted ratcheting rate and slope over stage II was found nearly the same and smaller than measured values.” in Chapter 6 to be less precise than the sentence of similar context in the previous chapter where the ratcheting rate was described as one-half of the rate of measured ratcheting data over stage II.

52. Authors's response: 

The sentence was omitted from Chapter 6.

53. Finally, since the Authors are not consistent in the naming (Neuber, Glinka, and H-S models vs Neuber, Glinka, and H-S rules), it should be more clearly written what models are mentioned in the last sentence: FE models or the Neuber, Glinka, and H-S models

53. Authors's response:

Naming is corrected according to the reviewer’s suggestion.

Reviewer 2 Report

The paper is interesting in terms of the methodology used. Before publication, some changes should be made.

The authors should re-organise Section 3, as it confuses methodology and morphology of the steel plates with the results. The two should be separated. 

-A diagram or schematic of the different configurations is desirable, as well as a photo of the test showing parts, elements, etc. 

-The nomenclature of the specimens is not clear, S3, S11, S16.... Please clarify this aspect

-Line 236. Please do not leave loose sentences between figures.

-In some parts of the text the reference to figures appears as figures 2. Please correct to Figures 2.

-Please, between numbers and units introduce a space, i.e. 0.40 mm

-The Y-axis in Figure 4 is chaotic. Please use numbers at the same scale and spacing.

-It is rare that the 4 loading situations coincide with their variations. Authors should clarify this aspect. Either they are not those deviation values or it is not well expressed.

Author Response

1. The paper is interesting in terms of the methodology used. Before publication, some changes should be made.

1. Authors's response:

Authors thank the reviewer for the encouraging comment.

2.The authors should re-organise Section 3, as it confuses methodology and morphology of the steel plates with the results. The two should be separated. 

2. Authors's response:

With a great respect to the reviewer comments, section 3 intends to hold testing conditions and ratcheting data for notched 1045 steel samples at various loading levels. These experimental procedure/data will support analysis and simulation conducted in the manuscript. A change in the structure will affect the entire manuscript structure.

3.A diagram or schematic of the different configurations is desirable, as well as a photo of the test showing parts, elements, etc. 

3. Authors's response:

This has been addressed as suggested b the reviewer.

4.The nomenclature of the specimens is not clear, S3, S11, S16.... Please clarify this aspect

4. Authors's response:

These specimens were taken from reference [8]. The reference is given in Table 1.

5.Line 236. Please do not leave loose sentences between figures.

5. Authors's response:

Editorial corrections were made as suggested by the reviewer.

6.In some parts of the text the reference to figures appears as figures 2. Please correct to Figures 2.

6. Authors's response:

Editorial revisions were made as suggested by the reviewer.

7. Please, between numbers and units introduce a space, i.e. 0.40 mm

7. Authors's response:

Editorial changes were made as suggested by the reviewer.

8.The Y-axis in Figure 4 is chaotic. Please use numbers at the same scale and spacing.

8. Authors's response:

The purpose of placing different scales was to show the entire diagrams in a single figure box for the sake of comparison. It has been changed according to the reviewer’s suggestion.

9. It is rare that the 4 loading situations coincide with their variations. Authors should clarify this aspect. Either they are not those deviation values or it is not well expressed.

9. Authors's response:

Editorial revisions were made as suggested by the reviewer.

Reviewer 3 Report

The paper must necessarily include nomenclature - all symbols, abbreviations and markings must be presented there - they are not fully explained in the text - already at the beginning of paragraph 2, not everything is explained and described. These are things that need to be made clear, especially in journals of renown Materials MDPI.

Authors must carefully follow the text of the paper.

Abstract must provide basic information about what is in the manuscript - briefly succinctly, without referring to any results - it must be shortened and corrected, without giving any symbols, abbreviations and literature references.

The word "sample" should not be used for the material used in testing and research - the word "specimen" is preferred, as stated in ASTM or BS standards.

Please do not use the word "work" in relation to scientific articles - preferred words "manuscript, paper, scientific article".

The literature review is correct.

The authors should work a bit on writing mathematical formulas - especially those with many brackets and powers - sometimes the formulas are a bit illegible, you need to think about their form - examples of formulas (16) and (17). In addition, the patterns merge with the text - proper spacing should be provided between them - this is not in the paper - that's why it is hard to read. Please correct it.

In some places in the paper, there should be no indentation of the text - paragraphs - the paper is perceived badly - such an example is line (150) - a paragraph, i.e. indentation of the first line, is unnecessary.

Formula (24) is an example of a correctly written mathematical formula. Please fix all formulas like this.

Numbering of the formulas - should be exactly in the horizontal axis of the entire mathematical formula - please change it - example of wrong numbering of the formula - formula (24).

Please, enrich paragraph 2 of the paper with figures from source manuscripts - these figures can be developed by the authors themselves on the basis of these manuscripts. This will enrich the peer-reviewed manuscript and properly illustrate the scope of the message presented.

The paper in paragraph 3 should include a technical drawing of the specime with all dimensions. Please complete the manuscript

It is a pity that the authors did not repeat the tests for the same geometry and the same load to check the convergence of the results. The fact is that the authors studied two different geometries and different loads. In my opinion, the research program lacks comparative studies:

- for the same geometry and the same load,

- for this different geometry and the same load.

The authors took the tests for the same geometry and different loads from other paper - it is a pity that they did not conduct them themselves and did not repeat these results. The analysis of these results seems correct.

The fourth paragraph lacks information about the accurate size of finite elements, the number of finite elements, the number of nodes in a finite element and the total number of nodes in the model. It should be specified what finite elements were used by the authors in the research (type, type of formulation in FEs), how many nodal these elements were, how many points of numerical integration were in the finite element, how many Jacobian points were there. The division of the model into finite elements is puzzling - the area in the close vicinity of the notch is densely divided, while the remaining area is quite poorly divided. We are dealing with a large disproportion in the size of finite elements. This is rather unacceptable and may lead to incorrect numerical estimates. The division in the direction of thickness is surprising - we see one layer of finite elements per 3mm - assuming that the real element at this thickness can be dominated by a plane stress state, such a calculation model certainly leads to incorrect results. The model presented by the authors would be acceptable if it would be advisable to compare it with a densely divided model and if the results of the model presented in the paper converge. The authors also do not write anything about the convergence of the results - it is unacceptable. The concept of convergence appears on the occasion of Figure 4 - which I do not understand at all - what the presented results can be related to - it is very questionable.

The third figure does not properly show the numerical model. This figure should show and describe the boundary conditions, axes of symmetry, places of load application and possible fastening. Please correct it.

What do the authors understand by the term "close agreement between stress-strain hysteresis loop"? Please provide the criterion for the "good agreement" condition - it does not currently result from the paper - see Figure 5.

The figures are illegible - it is impossible to identify in Figure 5 what is the result of FEM calculations and what is the experiment, which should be shown with proper citation of the literature, if the authors did not do new research.

Physical units in figures should be given after a space, preferably in square brackets.

The unit of stress is "MPa" - not "Mpa" - please correct this in some places in the paper, especially in the figures.

In general, the paper has potential, however, I currently believe that it should be successively corrected, supplemented and re-reviewed. I suggest a major revision.

Author Response

1.The paper must necessarily include nomenclature - all symbols, abbreviations and markings must be presented there - they are not fully explained in the text - already at the beginning of paragraph 2, not everything is explained and described. These are things that need to be made clear, especially in journals of renown Materials MDPI.

2.  Authors's response:

As suggested by the reviewer, a nomenclature has been added to the manuscript. Terms and variable were defined.

2.Abstract must provide basic information about what is in the manuscript - briefly succinctly, without referring to any results - it must be shortened and corrected, without giving any symbols, abbreviations and literature references.

 2. Authors's response:

Editorial revisions were made as suggested by the reviewer.

3.The word "sample" should not be used for the material used in testing and research - the word "specimen" is preferred, as stated in ASTM or BS standards.

3. Authors's response:

Editorial revisions were made as suggested by the reviewer.

4.Please do not use the word "work" in relation to scientific articles - preferred words "manuscript, paper, scientific article".

4. Authors's response:

Editorial revisions were made as suggested by the reviewer.

5.The authors should work a bit on writing mathematical formulas - especially those with many brackets and powers - sometimes the formulas are a bit illegible, you need to think about their form - examples of formulas (16) and (17). In addition, the patterns merge with the text - proper spacing should be provided between them - this is not in the paper - that's why it is hard to read. Please correct it.

5. Authors's response:

Revisions were made in writing mathematical expressions, as suggested by the reviewer.

6. In some places in the paper, there should be no indentation of the text - paragraphs - the paper is perceived badly - such an example is line (150) - a paragraph, i.e. indentation of the first line, is unnecessary.

 6. Authors's response:

Editorial revisions were made as suggested by the reviewer.

7. Formula (24) is an example of a correctly written mathematical formula. Please fix all formulas like this.

7. Authors's response:

Expressions were re-edited as suggested by the reviewer.

8.Numbering of the formulas - should be exactly in the horizontal axis of the entire mathematical formula - please change it - example of wrong numbering of the formula - formula (24).

 8. Authors's response:

Equation numbering was corrected as the reviewer suggested.

9. Please, enrich paragraph 2 of the paper with figures from source manuscripts - these figures can be developed by the authors themselves on the basis of these manuscripts. This will enrich the peer-reviewed manuscript and properly illustrate the scope of the message presented.

9. Authors's response:

Paragraph 2 in the manuscript provided the recent research on cyclic plasticity and ratcheting of materials at the presence of notch. This paragraph outlines the mission of this manuscript in employing kinematic hardening rules along with Neuber, Glinka and H-S models to assess local ratcheting at notch root of 1045 steel samples. It further highlights simulation of ratcheting at notch root by means of FE analysis and on the basis of Chaboche’s hardening rule.

10. The paper in paragraph 3 should include a technical drawing of the specimen with all dimensions.

10. Authors's response:

This has been addressed as suggested by the reviewer.

11. It is a pity that the authors did not repeat the tests for the same geometry and the same load to check the convergence of the results. The fact is that the authors studied two different geometries and different loads. In my opinion, the research program lacks comparative studies:

- for the same geometry and the same load,

- for this different geometry and the same load.

11. Authors's response:

A detailed discussion has been given right after ratcheting diagrams, as suggested by the reviewer.

12. The fourth paragraph lacks information about the accurate size of finite elements, the number of finite elements, the number of nodes in a finite element and the total number of nodes in the model. It should be specified what finite elements were used by the authors in the research (type, type of formulation in FEs), how many nodal these elements were, how many points of numerical integration were in the finite element, how many Jacobian points were there.

12.  Authors's response:

Editorial comments were addressed. Details of FE model/ analysis were provided in the manuscript as suggested by the reviewer.

13.The division of the model into finite elements is puzzling - the area in the close vicinity of the notch is densely divided, while the remaining area is quite poorly divided. We are dealing with a large disproportion in the size of finite elements. This is rather unacceptable and may lead to incorrect numerical estimates. The division in the direction of thickness is surprising - we see one layer of finite elements per 3mm - assuming that the real element at this thickness can be dominated by a plane stress state, such a calculation model certainly leads to incorrect results. The model presented by the authors would be acceptable if it would be advisable to compare it with a densely divided model and if the results of the model presented in the paper converge. The authors also do not write anything about the convergence of the results - it is unacceptable. The concept of convergence appears on the occasion of Figure 4 - which I do not understand at all - what the presented results can be related to - it is very questionable.

13. Authors's response:

Since only the element at the notch root was used to obtain the ratcheting data, the elements in the vicinity of the notch root played a more important role in the finite element analysis. For this purpose, convergence analysis was done by changing the size of the elements in the neighborhood of the notch root. This analysis was run over 100 cycles and the first, 20th, 50th, 70th, and 100th cycles had similar results. In other words, through several runs, over hundred cycles, convergence was consistently achieved for element size ranging between 0.15 mm and 0.40 mm. For this element range size ratcheting at the vicinity of notch root stayed nearly constant as presented in figure 5. However, according to the respected reviewer’s suggestion, the element’s size was also gradually reduced further from the notch root to 1 mm distances (illustrated in figure 4).No change in the ratcheting strain at the notch root was evidenced.

14.The third figure does not properly show the numerical model. This figure should show and describe the boundary conditions, axes of symmetry, places of load application and possible fastening. Please correct it.

14. Authors's response:

Editorial revisions were made to properly present numerical model as suggested by the reviewer.

15. What do the authors understand by the term "close agreement between stress-strain hysteresis loop"? Please provide the criterion for the "good agreement" condition - it does not currently result from the paper - see Figure 5.

15. Authors's response:

The close agreement term means the smallest difference with the experimental values and also maintaining the shape of the hysteresis loop. In the third diagram, there are many common points between the simulated loop and the experimental loop at both ends.

16. The figures are illegible - it is impossible to identify in Figure 5 what is the result of FEM calculations and what is the experiment, which should be shown with proper citation of the literature, if the authors did not do new research.

16. Authors's response:

Editorial revisions were made to further clarify the FE results as suggested by the reviewer.

17. Physical units in figures should be given after a space, preferably in square brackets.

17. Authors's response:

Editorial revisions were made as suggested by the reviewer.

18.The unit of stress is "MPa" - not "Mpa" - please correct this in some places in the paper, especially in the figures.

18. Authors's response:

Editorial revisions were made as suggested by the reviewer.

19. In general, the paper has potential, however, I currently believe that it should be successively corrected, supplemented and re-reviewed. I suggest a major revision.

19. Authors's response:

Authors appreciate the positive comments made by the reviewer.

Round 2

Reviewer 3 Report

The authors included almost all my corrections in the manuscript. I recommend the manuscript for publication.